# The Application of Phytohormones as Biostimulants in Corn Smut Infected Hungarian Sweet and Fodder Corn Hybrids

**DOI:** 10.3390/plants10091822

**Published:** 2021-09-01

**Authors:** Lóránt Szőke, Makoena Joyce Moloi, Gabriella Enikő Kovács, Györgyi Biró, László Radócz, Mária Takácsné Hájos, Béla Kovács, Dalma Rácz, Mátyás Danter, Brigitta Tóth

**Affiliations:** 1Faculty of Agricultural and Food Sciences and Environmental Management, Institute of Food Science, University of Debrecen, 138 Böszörményi St., 4032 Debrecen, Hungary; szoke.lorant@agr.unideb.hu (L.S.); kovacsb@agr.unideb.hu (B.K.); 2Department of Plant Sciences, Faculty of Natural and Agricultural Sciences, University of the Free State-Main Campus, P.O. Box 339, 9300 Bloemfontein, South Africa; MoloiMJ@ufs.ac.za; 3Faculty of Agricultural and Food Sciences and Environmental Management, Institute of Plant Protection, University of Debrecen, 138 Böszörményi St., 4032 Debrecen, Hungary; kovacs.gabriella@agr.unideb.hu (G.E.K.); ferencsikne.gyorgyi@agr.unideb.hu (G.B.); radocz@agr.unideb.hu (L.R.); 4Faculty of Agricultural and Food Sciences and Environmental Management, Institute of Horticultural Science, University of Debrecen, 138 Böszörményi St., 4032 Debrecen, Hungary; hajos@agr.unideb.hu; 5Institute of Land Use Technology and Regional Development, University of Debrecen, 138 Böszörményi St., 4032 Debrecen, Hungary; racz.dalma@agr.unideb.hu; 6Modern Application Platform Business Unit, VMware, Inc. 3401 Hillview Ave, Palo Alto, CA 94304, USA; mdanter@gmail.com

**Keywords:** antioxidant system, biostimulants, biotic stressor, chlorophyll, oxidative damage, stress response

## Abstract

The main goal of this research was to investigate the effects of corn smut (*Ustilago maydis* DC. Corda) infection on the morphological (plant height, and stem diameter), and biochemical parameters of *Zea mays* L. plants. The biochemical parameters included changes in the relative chlorophyll, malondialdehyde (MDA), and photosynthesis pigments’ contents, as well as the activities of antioxidant enzymes—ascorbate peroxidase (APX), guaiacol peroxidase (POD), and superoxide dismutase (SOD). The second aim of this study was to evaluate the impact of phytohormones (auxin, cytokinin, gibberellin, and ethylene) on corn smut-infected plants. The parameters were measured 7 and 11 days after corn smut infection (DACSI). Two hybrids were grown in a greenhouse, one fodder (Armagnac) and one a sweet corn (Desszert 73). The relative and the absolute amount of photosynthetic pigments were significantly lower in the infected plants in both hybrids 11 DACSI. Activities of the antioxidant enzymes and MDA content were higher in both infected hybrids. Auxin, cytokinin, and gibberellin application diminished the negative effects of the corn smut infection (CSI) in the sweet corn hybrid. Phytohormones i.e., auxin, gibberellin, and cytokinin can be a new method in protection against corn smut.

## 1. Introduction

Global warming, a consequence of climate change due to natural and anthropogenic causes, is becoming one of the biggest problems on Earth [1]. Connections between global warming and agriculture are well-studied [2]. A combination of abiotic and biotic stress factors, such as high temperature, water deficiency or over-supply, and biotic factors (e.g., pests, and pathogens), affect agricultural production negatively [3,4,5,6,7,8,9,10].

The outcomes of the effects of pathogens on the morphological and biochemical characteristics of host plants vary based on the pathogen [11,12,13,14,15]. One of the first reactions of host plants to biotic stress involves the overproduction of the reactive oxygen species (ROS). Pathogen-induced elevated ROS levels can contribute to the oxidative damage of cells, leading to lipid peroxidation and protein damage. *Pyricularia oryzae* infected wheat leaves had higher malondialdehyde (MDA) content compared to non-infected ones [16]. Similarly, Wu et al. [17] reported that MDA content was higher in rice leaves under *Rhizoctonia solani* infection. On the other hand, ROS could also have an important role in the resistance response of plants against pathogen infections [18]. Barna et al. [18] reported that increased antioxidant capacity and ROS levels have a significant role in systemic acquired resistance (SAR). The appropriate activity of the antioxidant enzymes is required to neutralize the harmful effects of ROS overproduction in host plants. The selected antioxidant enzymes in this study are playing a role in the transformation of free radicals into harmless metabolites [19]. Superoxide-dismutase (SOD) transforms superoxide into molecular oxygen and hydrogen peroxide. Ascorbate oxidase (APX) participates in ascorbate metabolism, using L-ascorbate and oxygen to produce dehydroascorbate and water. Peroxidase (POD) breaks down hydrogen peroxide producing molecular oxygen and water [20]. In some plants (*Coffea Arabica*, tobacco, and mung beans), pathogen infection upregulated the activities of ascorbate peroxidase (APX), superoxide dismutase (SOD), and guaiacol peroxidase (POD) [21,22,23,24]. 

Another aspect of tolerance to plant pathogens could involve phytohormones. Barna et al. [18] showed that hormones strongly affect the production of ROS and antioxidants, which can in turn influence the susceptibility or resistance of plants to pathogens. The roles of phytohormones such as ethylene, jasmonic acid, and salicylic acid in plants’ immune systems’ response to biotic stresses are widely examined for their role [25,26,27,28,29]. Additionally, auxins, gibberellins, and cytokinins are also described as resistance regulators [30,31,32]. Denancé et al. [31] stated that the application of phytohormones can enhance the resistance of host plants under biotic stress conditions. Manipulation of phytohormones’ homeostasis in crops can improve resistance response to biotic factors. Auxins are important for growth regulation [33] and pathogen resistance [34], e.g., auxin-aspartic acid plays a role in the resistance to a necrotrophic fungus, *Botrytis cinerea* [31]. Several studies have dealt with the role of ethylene in pathogen resistance. These found that the application of ethylene enhances the resistance or susceptibility of host plants. The effect depends on the type of pathogen and host plants [35,36]. Cytokinins are important in cell division and differentiation [37], chloroplast structure, senescence [38], and the reaction of host plants to biotic stressors [39,40]. Gibberellins can enhance or decrease the host plants’ resistance to phytopathogens, they play a role especially in the response of host plants to biotrophic and necrotrophic pathogens [41].

Corn smut (*Ustilago maydis* DC. Corda) has a major impact on corn production worldwide [42] and can infect corn at all phenological phases. The main symptoms of the disease are chlorosis, necrosis, tumor induction [43], and galls growth [44]. This pathogen prefers warm and dry weather conditions [45]. Therefore, global warming is favorable for its spread and virulence. There is no direct protection against the pathogen because the use of fungicides does not protect against corn smut infection. Only indirect protection methods can be used, e.g., avoid mechanical damage, appropriate pest-control management, and optimal nitrogen fertilization. On the other hand, *Ustilago maydis* can cause oxidative damage in host plants through enhanced free radical generation [46]. Hemetsberger et al. [47] reported that corn smut has a specific protein, called Pep1, which can block the activity of the peroxidase12 enzyme, resulting in the inhibition of oxidative burst as a response of the host to the corn smut infection. Pathogens activate the biosynthesis of hormones in plants [48]. Corn smut is a great example of a biotrophic fungus that has the ability to manipulate the host plant’s hormone balance. Bölker et al. [49] noticed that the hormone imbalance of the host caused by corn smut infection has a role in host infection. Research data shows that corn smut is able to produce cytokinins, abscisic acid [50], and auxins [51] to manipulate the host’s hormone balance. 

This study aimed to examine the effects of corn smut infection on the morphological (plant height, dry weight, and stem diameter), and biochemical (contents of photosynthetic pigments, malondialdehyde, and the activities of ascorbate peroxidase (APX), guaiacol peroxidase (POD), and superoxide dismutase (SOD)) parameters of maize. Such information will provide a clue on which of the biochemical mechanisms are important for tolerance to this pathogen, which could then be used for breeding purposes. In addition, this study investigated the effects of four phytohormones (auxin, cytokinin, gibberellin, and ethylene) on the physiological and biochemical parameters of corn smut infected maize, and their potential to provide instant protection. Additionally, the results of this study could be useful in hormone-based breeding programs to improve resistance to corn smut infection. The selected hormones’ concentrations applied in this experiment are higher than in plants because we wanted to examine the hormones’ effects, so their concentrations are unusually high. The morphological and biochemical characteristics of plants were measured twice, seven and eleven days after corn smut infection. Based on our previous experiment, which was conducted to observe the time of first symptoms of the CSI, the first symptoms (chlorosis, and tumor formation) of corn smut infection appeared 7 DACSI. This was the reason for the first sampling time. The second sampling time (11 DACSI) was chosen because sweet corn started to get necrosis symptoms 14 DACSI.

## 2. Results

Unless stated otherwise, the following comparisons were done: corn smut infected plants with non-infected plants (positive control); corn smut infected plants treated with phytohormones (cytokinin, gibberellin, auxin, and ethylene) to infected but hormone-free plants (negative control). Chlorophyll-a concentration was lower in the corn smut infected Desszert 73 (7 and 11 days after corn smut infection, or DACSI) and Armagnac (11 DACSI) relative to non-infected control. Infected Desszert 73 (11 DACSI) treated with cytokinin, gibberellin, and auxin, had significantly higher chlorophyll-a content compared to infected control. However, 7 DACSI, cytokinin was the only hormone that could significantly increase chlorophyll-a content in infected Desszert (to the level of the positive control), other hormones (gibberellin and auxin) did not have a significant impact (same levels as the negative control). Cytokinin treatment increased chlorophyll-a content of infected Armagnac, while the gibberellin and auxin treatments reduced it (7 DACSI). Treatment of infected Desszert 73 and Armagnac hybrids with ethylene substantially reduced the chlorophyll-a content at both sampling times (Figure 1).

The infection had no effect on the chlorophyll-b content 7 DACSI for both hybrids. However, this changed 11 DACSI in the infected plants where both hybrids had lower chlorophyll-b contents. As a result of three hormone treatments (cytokinin, gibberellin, and auxin), the amount of chlorophyll-b for infected Desszert 73 hybrid increased at both sampling times. In contrast, only cytokinin was effective in inducing an increase in the chlorophyll-b content (11 DACSI) while gibberellin and auxin either reduced its accumulation or were not effective (same letter with the negative control) at both sampling times in infected Armagnac. Infected Armagnac and Desszert 73 treated with ethylene had significantly reduced chlorophyll-b content at both sampling times (Figure 2).

The infection did not affect carotenoid content 7 DACSI but significantly lowered it later on (11 DACSI) for both hybrids. Cytokinin, gibberellin, and auxin treatments substantially increased the carotenoid content of infected Desszert 73 (11 DACSI). For infected Armagnac, only cytokinin and gibberellin were effective in inducing carotenoid accumulation (11 DACSI). For both infected hybrids, treatment with auxin led to significantly reduced carotenoid accumulation for 7 DACSI, and 11 DACSI the amount of carotenoid increased only for Desszert 73. Cytokinin did not have a significant effect for both hybrids 7 DACSI, while gibberellin reduced carotenoid in Armagnac 7 DACSI. Treatment of the two examined infected hybrids with ethylene decreased the carotenoid content significantly at both sampling times (Figure 3).

As the result of the corn smut infection, both hybrids had higher MDA (malondialdehyde) content 7 and 11 DACSI. Treatment of infected Armagnac with cytokinin, gibberellin, and auxin had no significant effect on MDA production 7 DACSI. For Desszert 73, cytokinin and auxin reduced MDA production 7 DACSI. Treatment of infected plants with ethylene increased the amount of MDA for both hybrids at both sampling times (Figure 4).

Infected plants had increased POD and APX activities for both hybrids 7 and 11 DACSI compared to the control. The cytokinin, gibberellin, and auxin treatment reduced the POD and APX activities in the infected plants for the two hybrids at both sampling times. Infected plants treated with ethylene showed the highest POD and APX activities compared to the control plants at both sampling times (Figure 5 and Figure 6).

Corn smut infection induced significantly higher SOD activity at both sampling times for the two hybrids compared to the non-infected control. Cytokinin, gibberellin, and auxin treatments reduced the SOD activity in the infected plants for both hybrids 7 and 11 DACSI. Ethylene treatment significantly increased the SOD activity in Armagnac hybrid 11 DACSI relative to the CSI treatment (Figure 7).

Desszert 73 was the only hybrid with significantly reduced plant height after corn smut infection (7 DACSI). Cytokinin, gibberellin, and auxin treatment induced plant height in the infected (7 DACSI) Armagnac hybrid. Infected Armagnac hybrid treated with cytokinin and ethylene had significantly reduced plant height 11 DACSI. Ethylene treatment significantly reduced plant height at both sampling times except for Armagnac (7 DACSI). The infected plants had lower plant height than the control for Desszert 73 hybrid at both sampling times. This observation was similar in the Armagnac at 11 DACSI (Figure 8).

The corn smut infection significantly decreased the shoot dry weight for Desszert 73 (7 and 11 DACSI) and Armagnac (11 DACSI). Infected plants (11 DACSI) treated with gibberellin had higher shoot dry weight whereas the ones treated with cytokinin and auxin had reduced shoot dry weight. The auxin treatment significantly increased the shoot dry weight for infected Armagnac (7 DACSI). Ethylene treatment led to the lowest shoot dry weight in the infected hybrids at both sampling times (Figure 9).

The corn smut infected Desszert 73 had thicker stems 7 DACSI compared to control. Gibberellin treatment increased stem diameter in infected Armagnac and decreased stem diameter in infected Desszert 73 11 DACSI. Infected plants treated with ethylene had increased stem diameter for both hybrids at the two sampling times (Figure 10).

Table 1 demonstrates the average values of two hybrids. As compared to the control plants, the infected plants had lower contents of chlorophyll-a, chlorophyll-b, and carotenoid, shorter plant height, smaller stem diameter, higher MDA content, and higher activities of POD, APX, and SOD. The positive effects of plant regulators (cytokinin, gibberellin, and auxin) were mainly observed for MDA (decreased), and reduced APX, POD, and SOD activities compared to the hormone-free infected plants. Ethylene treatment negatively affected the above-mentioned parameters in the infected plants compared to the control plants.

Armagnac had higher chlorophyll-b and plant height compared to Desszert 73. The Desszert 73 hybrid had a thicker stem and higher MDA content and APX activity than the Armagnac hybrid. Both varieties being hybrids had no effect on the chlorophyll-a content, carotenoids contents, shoot dry weight, POD, and SOD activity (Table 2).

## 3. Discussion

According to the results, longer exposure to corn smut infection triggers the breakdown of the chlorophyll pigments. Corn smut infection significantly reduced the amount of photosynthetic pigments in both hybrids 11 DACSI (Figure 1, Figure 2 and Figure 3). Chlorosis is one of the typical symptoms of corn smut infection during the vegetative stage [52]. Other pathogens also diminish the chlorophyll content of the host plants. Chávez-Arias et al. [53] stated that the *Fusarium oxysporum f.* sp. *physali* decreased the chlorophyll content of Cape gooseberry (*Physalis peruviana* L.). Gonçalves et al. [54] declared that the chlorophyll-a and chlorophyll-b contents were lower in the sugarcane leaves of plants infected by the sugarcane mosaic virus (SMV) when compared to non-infected plants. Additionally, Kyrychenko [55] measured lower amounts of photosynthetic pigments in beans infected by the bean yellow mosaic virus (BYMV). Additionally, the investigated fodder and sweet corn hybrids had different responses to phytohormones application with respect to their photosynthetic pigments content. Cytokinins (CKs) play a role in chlorophyll synthesis, chloroplast ultrastructure, and chloroplast differentiation [56]. Dobránszki and Mendler-Drienyovszki [57] stated that CK treatment has an impact on chlorophyll-a, and b content in vitro apple leaves. Its effect depends on the applied concentration and form of CKs. Additionally, the application of phenylacetic and α-naphtylacetic acid reduced the chlorophyll and carotenoid contents in *Wolffia arrhiza* [58]. The impacts of phytohormones on plants’ chlorophyll degradation were examined by Misra and Biswal [59]. They stated that auxin and kinetin reduce chlorophyll degradation, while gibberellin (GA) stimulates the degradation. Research studies have shown that indol-acetic acid (IAA) and naphthaleneacetic acid (NAA) treatments caused chlorophyll loss in tobacco [60], and in lettuce [61]. Ethylene had a similar effect on chlorophyll. Ceusters and Van de Poel [62] published that the application of ethylene causes the degradation of chlorophyll through induced chlorosis. 

One possible way to evaluate the effects of stress factors is the level of lipid peroxidation, expressed in malondialdehyde (MDA) content [63]. In this research, CSI increased the MDA content of the infected plants in both hybrids at both sampling times (Figure 4). The investigated maize hybrids could not tolerate the effect of the corn smut infection, because the MDA content was significantly higher in the infected plants than in the control plants. High MDA content causes membrane damage and cell death [64]. More research also stated the negative effects of plant diseases on the MDA content of the host plants. According to Monnazah et al. [65], the amount of MDA was higher in the *Sclerotinia sclerotiorum* infected sunflower plants. *Colletotrichum gloeosporioides* infection also increased the MDA content in the cowpea plants [66] compared to non-infected plants. Phytohormones play a role in the stimulation of fatty acid biosynthesis, mitigate the negative effects of oxidative burst, and increase lipid production [67,68]. Sivaramakrishnan and Incharoernsakdi [69] noticed that MDA content was significantly higher after IAA and GA treatments in *Chlorella* sp. Higher MDA content was observed at IAA relative to GA with a concentration of 1 mM. They concluded that the application of IAA induces the generation of free radicals, which causes higher MDA content and cell damage. The MDA content of CSI and auxin treated fodder corn was significantly higher at 7 DACSI, compared to CSI treatment. While, in sweet corn under the same treatment, the MDA content was significantly lower 7 and 11 DACSI, relative to CSI plants. Treatment of corn smut infected plants with ethylene increased MDA content at both sampling times in both hybrids in this study while treatment with auxin, CK, and GA lowered the MDA 11 DACSI for both hybrids. This was in disagreement with the findings of Nazir et al. [70], who showed that phytohormones such as GA, kinetin, and jasmonic acid increased MDA production significantly. However, in line with our study, Yu et al. [71] stated that phytohormones, like GA, control the inner metabolic pathways because they diminish oxidative stress, reduce the rate of lipid peroxidation, and growth of cells. This shows that although CSI induced lipid peroxidation, the application of cytokinin, GA, and auxin increased the protection of the plants by minimizing MDA production. Therefore, the resistance or susceptibility of the host plant for plant pathogen infections can be induced by exogenous hormone treatments [41]. 

APX, SOD, and POD enzymes play an important role in the defense mechanisms of the host plant against biotic stressors such as plant pathogens. Plants try to compensate for the negative effects of stresses when the activities of these enzymes increase [72]. Corn smut infection induced higher POD, APX, and SOD activities in both hybrids at both sampling times in this study (Figure 5, Figure 6 and Figure 7). In addition, POD, APX, and SOD activities were lower in CSI plants when treated with auxin, GA, and CK in both hybrids at both sampling times. Ethylene treated and CSI plants had the highest antioxidant enzyme activities in this study. After the infection, the host plants produced a burst of H_2_O_2_ and ROS, which are important in oxidative stress signaling and cell death [73]. The antioxidant enzymes like APX, SOD, and POD can reduce the H_2_O_2_ and ROS accumulation [74]. In this experiment, maize plants reacted to corn smut infection in order to decrease free radical accumulation by increasing their enzymes’ activities. However, the increased enzyme activities did not reduce the MDA content of the host plant. The increased antioxidant enzymes activities, as the indicator of oxidative stress, are associated with reduced protein content [75]. Zehra et al. [76] stated, that *Fusarium oxysporum f.* sp. *lyopersici* increased the activity of APX in tomato plants. Fimognari et al. [77] published, that the *Pseudomonas syringae py. tabaci* reduced, while *Pseudomonas syringae pv. phaseolicola* increased the activity of APX in the *Nicotiana tabacum*. In addition, higher SOD activity was measured in *Botrytis cinerea* infected tomato leaves [78]. Paranidharan et al. [79] stated that the SOD activity was higher in rice plants due to the *Rhizoctonia solani* infection. The negative effects of phytopathogens on the POD activity of host plants are also well documented. According to Lee et al. [80], *Puccinia triticina* increased the POD activity of *Aegilops tauschii*, and *Fusarium verticillioides* infected maize plants also had higher POD activity [81] compared to the non-infected plants. 

The connection between the corn smut infection and the host plant’s morphological parameters was also studied in this research. The infection reduced the corn plants’ shoot dry weight of Desszert 73 at both sampling times and for Armagnac, at 11 DACSI (Figure 10). The infected plants had a thicker stem and lower plant height in the case of Desszert 73, at 7 DACSI (Figure 9 and Figure 10). The infection did not affect the above-mentioned parameters for the Armagnac hybrid. It would seem, therefore, that the Armagnac hybrid could tolerate the effect of CSI. However, the biochemical parameters (MDA content, and antioxidant enzymes activities) were influenced negatively by the infection. So it seems, that the fodder corn tolerance against corn smut was not detectable by measuring biochemical parameters. The negative effects can be inconspicuous on the morphological parameters, but the infection decreased host plant immunity and this was observed in this experiment. Researchers studied the effects of *Fusarium* spp. on the height of host plants. They found that the pathogens reduced the heights of tomato and brown nightshade plants [82,83]. According to Robert et al. [84] and Tahir et al. [85], *Septoria tritici*, and *Ralstonia solanacearum* infections did not decrease the dry weight of wheat and tobacco plants. In contrast, Prasch and Sonnewald [86] showed that the Turnip mosaic virus (*TuMV*) decreased the dry weight of *Arabidopsis thaliana*. The corn smut infected maize plants had thicker stems than the control plants [87]. So it seems that the effects of plant diseases on the host plant morphological parameters depend on the plant pathogen type and the host plant. The exogenous GA application had a positive effect on the plant height 7 DACSI, but the infection decreased the growth effect of the GA and the GA treated infected plants did not grow significantly more than the hormone-free infected plants after this period. So the infection had a large pathogenicity capacity, this is why the effects of GA were not detected 11 DACSI on the morphological parameters of the host plant. Additionally, the negative effects of rice dwarf mosaic virus were reduced by GA application because Zhu et al. [88] measured higher plant height compared to the hormone-free treated rice plant. Tanaka et al. [89] showed that probenazole-inducible protein (which was accountable for the resistance against the *Pyrcularia grisea*) was induced by the GA treatment in rice plants.

According to Argueso et al. [90] the susceptibility of the *Arabidopsis thaliana* host plant to the *Hyaloperonospora arabidopsidis* depended on the concentration of CK because a higher concentration of cytokinin increased, lower concentration of CK decreased the resistance of the host plant. It seems that GA can reduce the effects of biotic stresses, and reduce the antioxidant enzymes’ activities. The sensitivity for the *Agrobacterium tumefaciens* was high in the *Arabidopsis thaliana* due to the exogenous CK treatment [91]. 

After infection, the susceptible host plants produced some “negative” hormones such as ethylene, which weaken host plants [92]. If the levels of phytohormones are manipulated by adding other hormones such as auxin, the resistance can increase. Contrarily, the negative effects of auxin were also observed in Armagnac in some cases. Armagnac is a corn smut tolerant hybrid. Therefore, this needs further investigation to analyze the role of auxin in host plant-plant pathogen interaction when the host plant is resistant against or susceptible to this plant pathogen. Ethylene can increase the susceptibility or resistance of the host plant [36]. But the effects of ethylene in the infection cycle depend on the host plant, plant disease, and environmental factors [93]. Other researchers observed different connections between the effects of exogenous ethylene and plant diseases. Veselova et al. [94] concluded that the exogenous ethylene treatment increased the susceptibility of the wheat plants to the *Staganospora nodorum* pathogen, and aggravated the symptoms on the wheat leaves. In contrast with this, the ethylene hormone improved the resistance of the host plants (grapes and tomatoes) against the *Botrytis cinerea* pathogen [26,95]. 

Based on literary data, special receptors used for pattern recognition are located in membranes that act as the plants’ immune system response to fungal infection [96]. The activation of pattern-triggered immunity includes the triggers of reactive oxygen species (ROS), cytosolic ion-flux changes, calcium-dependent or mitogen-activated protein kinase cascade activation, reinforcement of physical barriers, and the production of several defense-related molecules [96,97] such as phytohormones [30,98,99]. Additionally, plant fungi are also able to synthesize phytohormones and regulate or capsize the hormone balance of the host plant [100]. Turian and Hamilton [101] reported high indole-3-acetic acid (IAA) content in tumors triggered by corn smut infection. This fungus uses tryptophan to produce IAA [51]. Reineke et al. [51] stated that three genes—two IAA dehydrogenases and one transaminase—reduce the IAA production of corn smut under controlled axenic conditions. However, the tumor formation was unchanged and the amount of IAA produced was identical to in wild-type infections. Based on this, the IAA that is synthesized by *Ustilago maydis* does not affect IAA levels in the tumor tissue or inducing the tumor [51]. Data of this study found that NAA reduced the generated MDA content during lipid peroxidation, and reduced the antioxidant enzymes’ activities in CSI and auxin treated plants relative to CSI plants. These data suggest that the exogenous application of NAA inhibits tumor formation, and mitigates the negative impacts of CSI on maize. The responsible gene for the biosynthesis of CKs is identified in several phytopathogens, so researchers suggested that pathogens, as well as fungi, produce them to aid the infection and disease [100,102]. The results of this study show that the activities of antioxidant enzymes such as SOD, POD, and APX (as the indicators of stress conditions triggered by corn smut) significantly declined relative to the CSI plants when CSI plants were exogenously treated with CK. This suggests that the over-application of CK compared to the *Ustilago maydis* produced amount itself, can mitigate the negative effects of the pathogen on the measured antioxidant enzymes. Research data of Bruce et al. [103] show that corn smut mainly produces cis isomers of Z-type CKs, which are accumulated in *Ustilago maydis* infected maize and tumor tissues. Maize plants contain CK receptors that respond to cisZ-type CKs and these accumulate during the corn smut infection [50,103]. Gibberellins are mainly examined in *Gramineae* pathogen phytosystems because of their biosynthesis by the *Fusarium fujikuroi* fungus. The gene which is responsible for GA synthesis is limited to different *Fusarium* varieties and the ability to produce GA is only found in *F. fujikuroi.* Research data show that the rice mutant which is absent of the GA inactivating enzyme is more susceptible to *M. oryzae* relative to the resistant mutant where GA-oxidase plays a role in GA biosynthesis [104]. Qin et al. [105] confirmed that GA pathways have a beneficial function in the *M. oryzae* rice phytosystem. Accordingly, the data of this study also confirm that the exogenous application of GA increases the resistance of maize to corn smut infection. De Vleesschauwer et al. [106] examined the application of ethephon as an ethylene-releasing regulator in the resistance of rice to *Cochliobolus miyabeanus.* Van Bockhaven et al. [107] found that ethylene is used by *C. miyabeanus* to forward virulence. They found rice sensitive to ethylene to be less resistant to that fungus than the non-ethylene sensitive rice. In addition to this, the ET produced by *C. miyabeanus* constituted the most significant source of ET in the tissues affected by the infection [107]. The data in this study also confirm that the exogenous application of an ethylene-based plant regulator enhances the pathogenicity of *Ustilago maydis*. According to current knowledge, only a few bio-stressors utilize the hormonal pathways of plants to their advantage. The reason that some fungi use proteins to synthesize phytohormones, and others modify the host plants’ hormone balance by producing phytohormones needs more study [108].

## 4. Materials and Methods

### 4.1. Growing of Plants and Applied Treatments

Test plants (*Zea mays* cv. Desszert 73, and Armagnac) were grown in a greenhouse to ensure optimal and constant environmental factors for the experiment. Maize plants were planted in peat in PVC tubes. Optimal temperatures for growth varied between 32 °C (daytime) and 25 °C (night), the humidity was kept above 40% (45–55%). Three plants were grown per tube, and one week after planting only well-developed maize plants were separated. The plants received 1 L of water/plant/day.

Six different treatments and ten maize plants per treatment (60 plants) were used including control, corn smut infected plants (CSI), and infected plants supplemented with phytohormones (cytokinin, gibberellin, auxin, and ethylene were used separately). Fungal infection and hormone treatments of corn plants were conducted simultaneously, at the four–five leaf phenology stage. The specific hormones and their concentrations were 2 × 10^−4^ M kinetin (cytokinin), 10^−3^ M gibberellin (GA_3_), 2 × 10^−3^ naphthaleneacetic acid (NAA, as auxin), and 1% ethrel (ethylene). Two ml fungal sporidium suspension (10,000 sporidium number/mL) was injected between the second and third nodes, and 1 mL of each hormone was also injected. Five plants from each treatment were selected to examine morphology and physiology parameters. All of the measurements were done 7 and 11 days after the corn smut infection (DACSI). The applied corn smut inoculum was created by the method of Szőke et al. [87].

### 4.2. Photosynthetic Pigment Quantification

Individual photosynthetic pigments (mg g^−1^ /fresh weight) were determined following the method described by Moran and Porath [109], and calculated by Welburn [110]. For measurement, a 50 mg fresh tissue sample was taken from the fourth leaf and dissolved in 5 mL N, N- dimethylformamide at 4 °C for 72 h. Spectrophotometric absorbance of the extracts was measured at 480 nm, 647 nm, and 664 nm wavelengths by using Nicolet Evolution 300 UV-Vis Spectrometer (Thermo Fisher Scientific, Waltham, MA, USA).

### 4.3. Morphological Parameters

The stem diameter between the second and third nodes was determined by using a slide caliper.

Plant height was measured from the peat surface to the origin of the youngest leaf at the top.

To determine the shoot dry weight, shoots were collected and dried at 65 °C for three days, then dried samples were measured by Ohaus AX223 (OHAUS Corporation, Parsippany, NJ, USA) analytical scale.

### 4.4. Determination of the Level of Lipid Peroxidation

The method for measuring the level of lipid peroxidation used in this experiment was described by Heath and Packer [111]. For measurement, 0.1 g leaf samples were ground with liquid nitrogen and homogenized in 1 mL solution containing 0.25% (*w/v*) thiobarbituric acid (TBA) and 10% (*w/v*) trichloroacetic acid (TCA). Samples were transferred to Eppendorf tubes (WVR International, PA, USA) and centrifuged at 10,800× *g* for 25 min at 4 °C. The supernatant (0.2 mL) was transferred to a clean Eppendorf tube which contained 0.8 mL solution of 0.5% (*w/v*) TBA and 20% (*w/v*) TCA. This mixture was heated to 95 °C for 30 min by using a thermoshaker (Bioshan TS-100) and then cooled rapidly on ice. Spectrophotometric absorbance (Nicolet Evolution 300 UV-Vis Spectrometer, Thermo Fisher Scientific, Waltham, MA, USA) was measured at 532 and 600 nm. The MDA content calculation was accomplished by using the extinction coefficient of 155 mM^−1^ cm^−1^.

### 4.5. Activity of Antioxidant Enzymes

The leaf samples were prepared using the method described in Pukacka and Ratahczak [112] and were used to measure activities of ascorbate peroxidase (APX), and guaiacol peroxidase (POD). Leaf samples (0.2 g) were homogenized to a fine paste in 1 mL 50 mM potassium phosphate buffer (pH 7.0) containing 2% (*w/v*) polyvinylpyrrolidone (PVP), 1 mM ascorbate, 0.1% (*v/v*) Triton X-100, and 1 mM ethylenediaminetetraacetic acid (EDTA). To separate the components of the mixture the samples were centrifuged at 15,000× *g* for 20 min at 4 °C. The obtained supernatant was transferred into a sterile Eppendorf tube and stored on ice until processing.

For the APX assay, the method of Mishra et al. [113] was followed with modifications. The assay mixture (1 mL) consisted of the following ingredients: 550 µL of 50 mM potassium phosphate buffer (pH 7.0), 200 µL H_2_O_2_ (0.1 mM), 150 µL sodium ascorbate (0.5 mM), 50 µL EDTA (0.1 mM EDTA) and 50 µL sample extract. Due to ascorbate oxidation, decreasing absorbance was measured at 290 nm for 5 min at 20 °C compared to a blank that contained phosphate buffer in place of the enzyme extract. The extinction coefficient of 2.8 mM^−1^ cm^−1^ was used.

The POD assay as proposed by Zieslin and Ben-Zaken [114] was adopted for evaluating the activity of guaiacol peroxidase. A mixture was composed of 50 µL 0.2 M H_2_O_2_, 100 µL 50 mM guaiacol, 340 µL distilled water, 490 µL 80 mM phosphate buffer (pH 5.5), and 20 µL sample extract. The POD activity was determined based on the generated concentration of tetraguaiacol. The absorbance of the reaction compound was read at 470 nm for 3 min at 30 °C. The sample extract was replaced by 50 mM phosphate buffer in the blank. For tetraguaiacol concentration estimation, the extinction coefficient of 26.6 mM^−1^ cm^−1^ was used.

The activity of superoxide dismutase (SOD) was measured based on the 50% inhibition of the reduction of nitroblue tetrazolium (NBT) and measured at 560 nm as proposed by Giannopolities and Ries [115], and Beyer and Fridovich [116]. Leaf samples (0.4 g) were homogenized in 4 mL 50 mM phosphate buffer (pH 7.8) containing 0.1 mM EDTA, 1% (*w/v*) PVP, and 1 mM phenylmethanesulfonyl fluoride (PMSF). Samples were centrifuged at 10,000× *g* for 15 min at 4 °C.

The method of Bradford [117] was adopted to measure the protein content of sample extracts. The activities of antioxidant enzymes and protein content were quantified at 7 and 11 DACSI.

### 4.6. Statistical Analysis

The statistical analysis was performed by IBM SPSS Statistics 25 (Armonk, NY, USA) software. Kolmogorov—Smirnov and Shapiro—Wilk tests [118] were used for data normality test and, the results were investigated by one-way ANOVA [119], the means were compared by Tukey HSD test [120]. Significance was marked by using lower case letters (a, b, c, and d) in the manuscript. The number of replications was five per treatment per parameter for statistical analysis.

## Figures and Tables

**Figure 1 plants-10-01822-f001:**
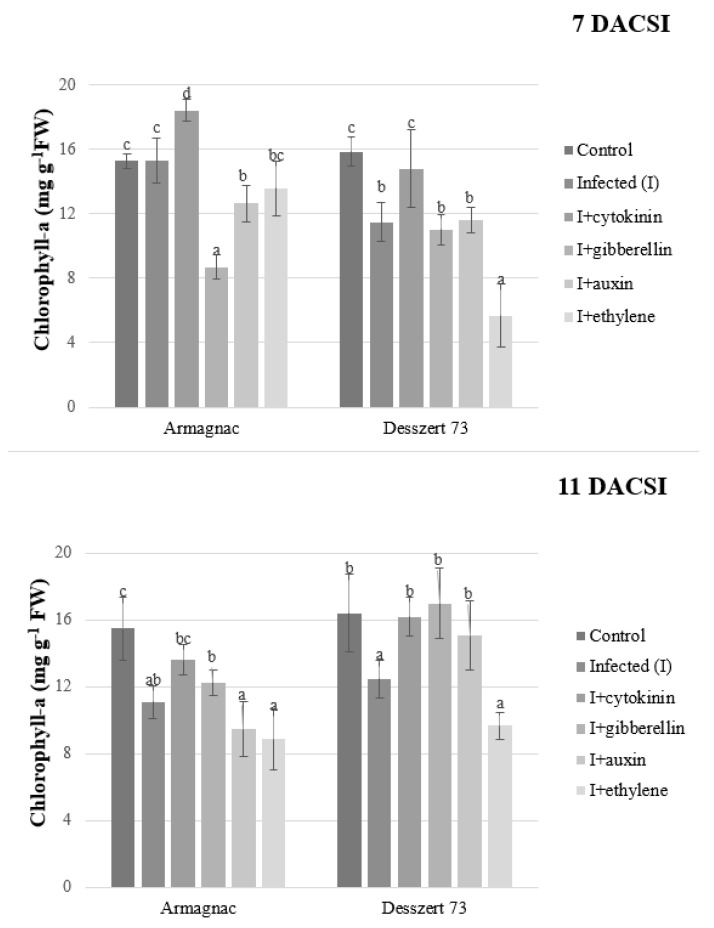
The chlorophyll-a content (mean ± SD, *n* = 5) of Armagnac and Desszert 73 hybrids (7 and 11 DACSI). Lower case letters (a, b, c, and d) show significant differences among the treatments based on the Tukey HSD test (*p* < 0.05).

**Figure 2 plants-10-01822-f002:**
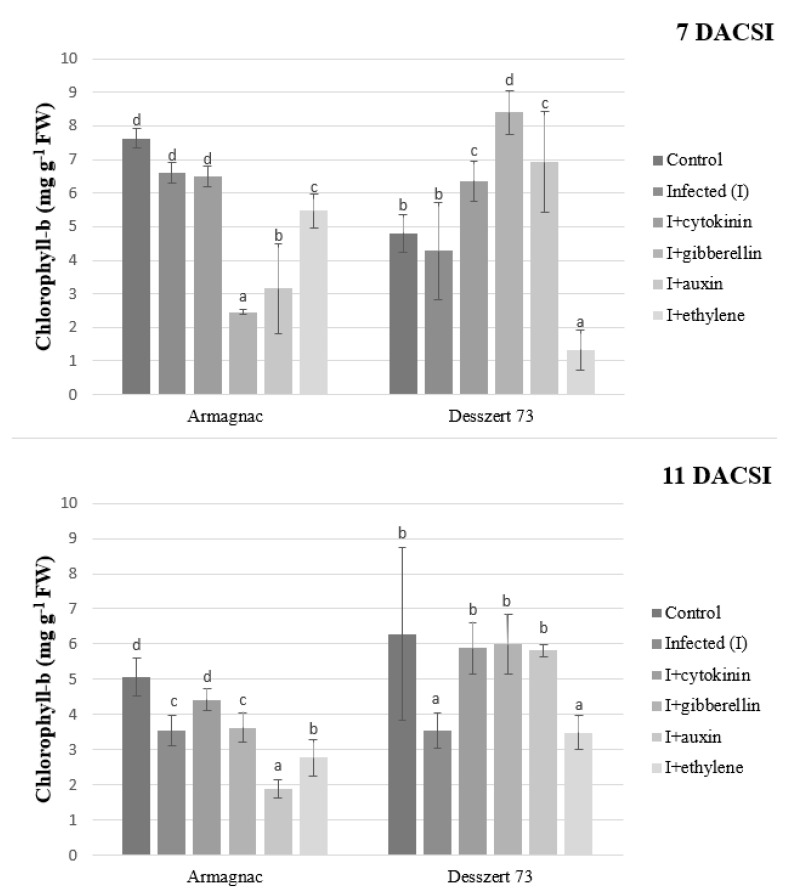
The chlorophyll-b content (mean ± SD, *n* = 5) of Armagnac and Desszert 73 hybrids (7 and 11 DACSI). Lower case letters (a, b, c, and d) show significant differences among the treatments based on the Tukey HSD test (*p* < 0.05).

**Figure 3 plants-10-01822-f003:**
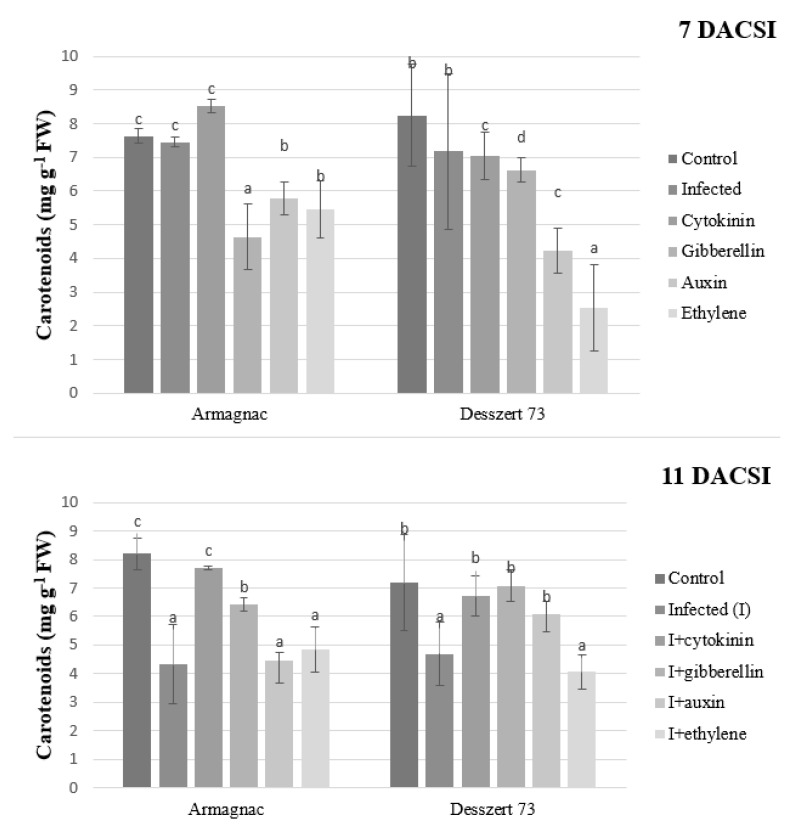
The carotenoid content (mean ± SD, *n* = 5) of Armagnac and Desszert 73 hybrids (7 and 11 DACSI). Lower case letters (a, b, c, and d) show significant differences among the treatments based on the Tukey HSD test (*p* < 0.05).

**Figure 4 plants-10-01822-f004:**
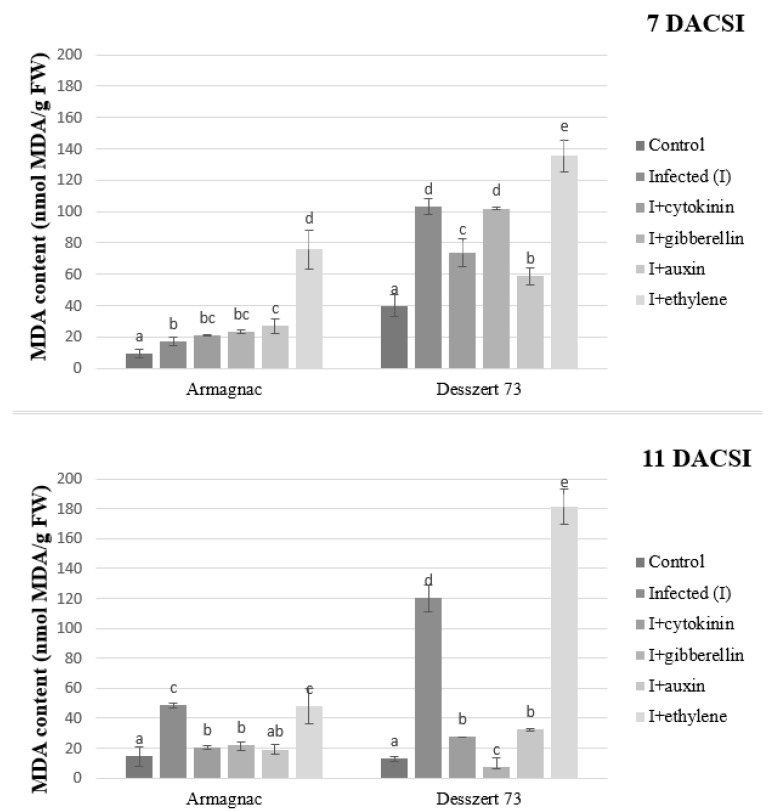
The MDA content (mean ± SD, *n* = 5) of Armagnac and Desszert 73 hybrids (7 and 11 DACSI). Lower case letters (a, b, c, d, and e) show significant differences among the treatments based on the Tukey HSD test (*p* < 0.05).

**Figure 5 plants-10-01822-f005:**
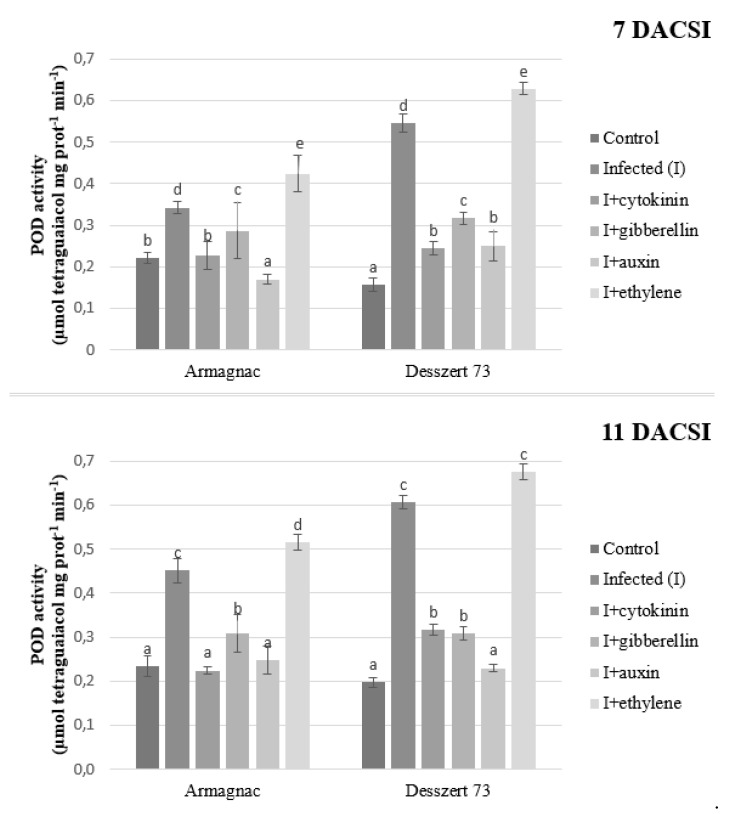
The peroxidase (POD) activity (mean ± SD, *n* = 5) of Armagnac and Desszert 73 hybrids (7 and 11 DACSI). Lower case letters (a, b, c, d and e) show significant differences among the treatments based on the Tukey HSD test (*p* < 0.05).

**Figure 6 plants-10-01822-f006:**
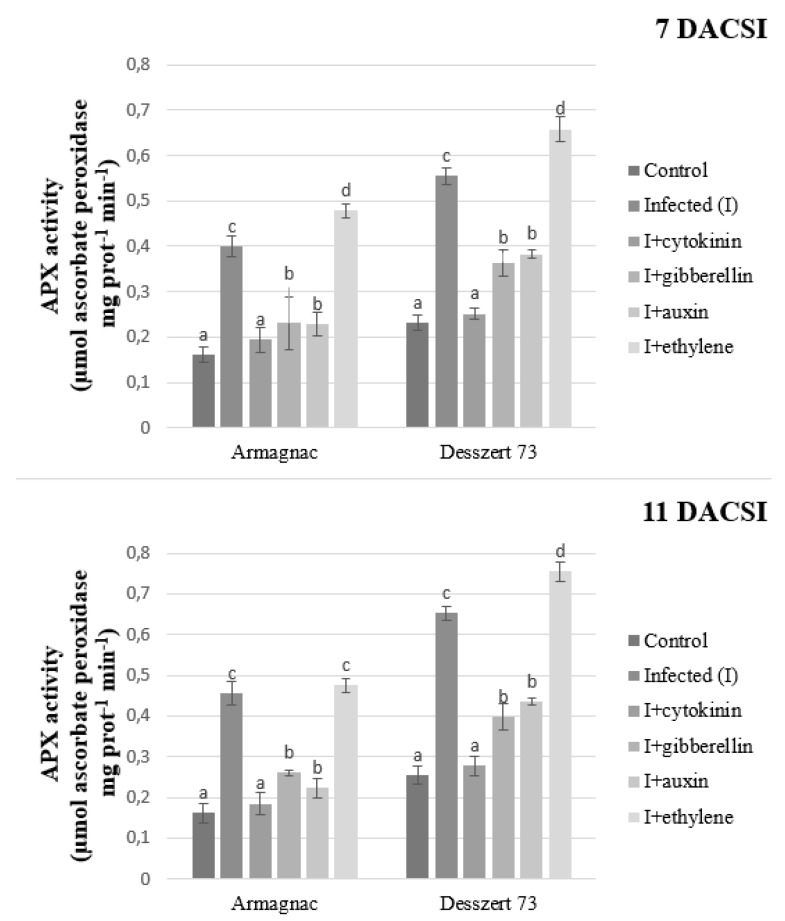
The APX activity (mean ± SD, *n* = 5) of Armagnac and Desszert 73 hybrids (7 and 11 DACSI). Lower case letters (a, b, c, and d) show significant differences among the treatments based on the Tukey HSD test (*p* < 0.05).

**Figure 7 plants-10-01822-f007:**
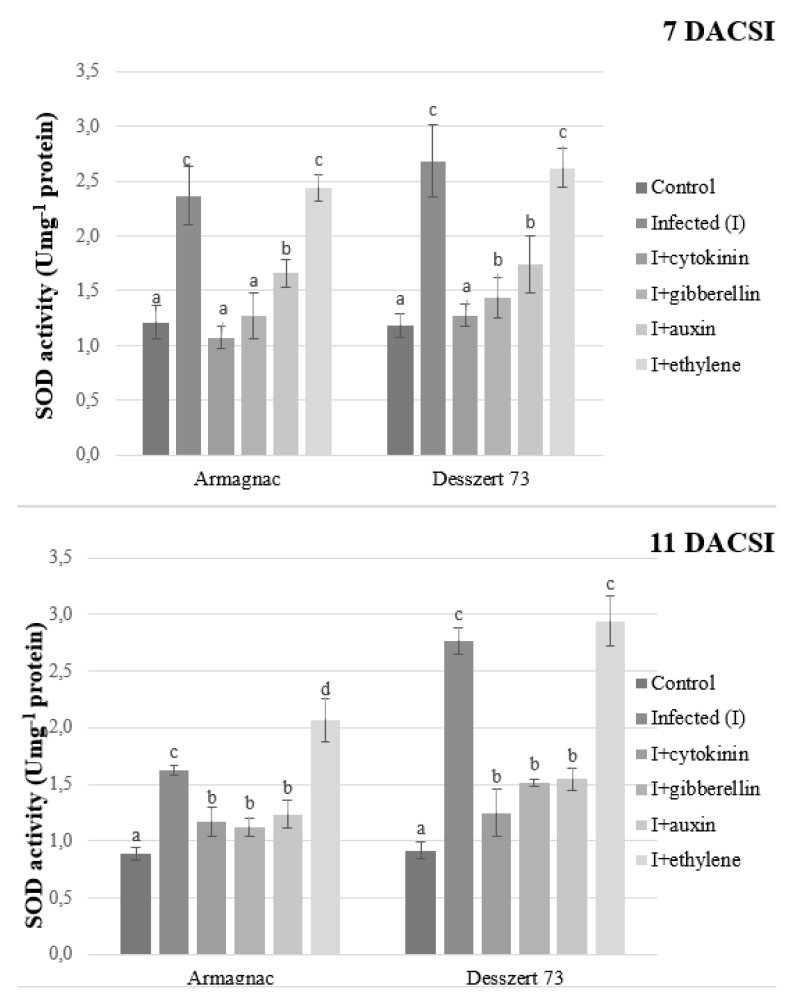
The SOD activity (U mg^−1^ protein, mean ± SD. *n* = 5) of Armagnac and Desszert 73 hybrids (seven and 11 DACSI). Lower case letters (a, b, c and d) show significant differences among the treatments based on the Tukey HSD test (*p* < 0.05).

**Figure 8 plants-10-01822-f008:**
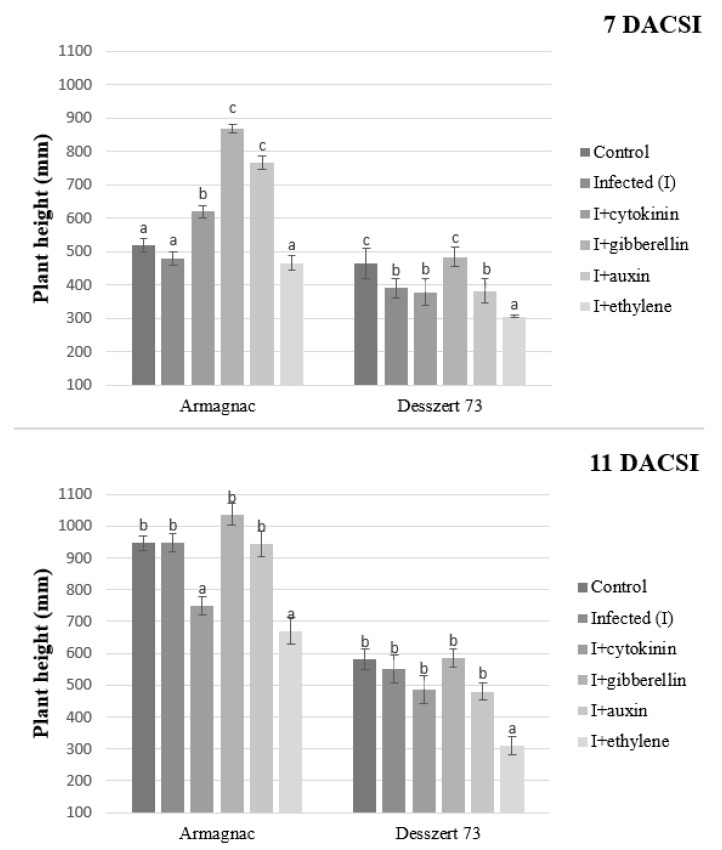
The plant height (mean ± SD, *n* = 5) of Armagnac and Desszert 73 hybrids (7 and 11 DACSI). Lower case letters (a, b, and c) show significant differences among the treatments based on the Tukey HSD test (*p* < 0.05).

**Figure 9 plants-10-01822-f009:**
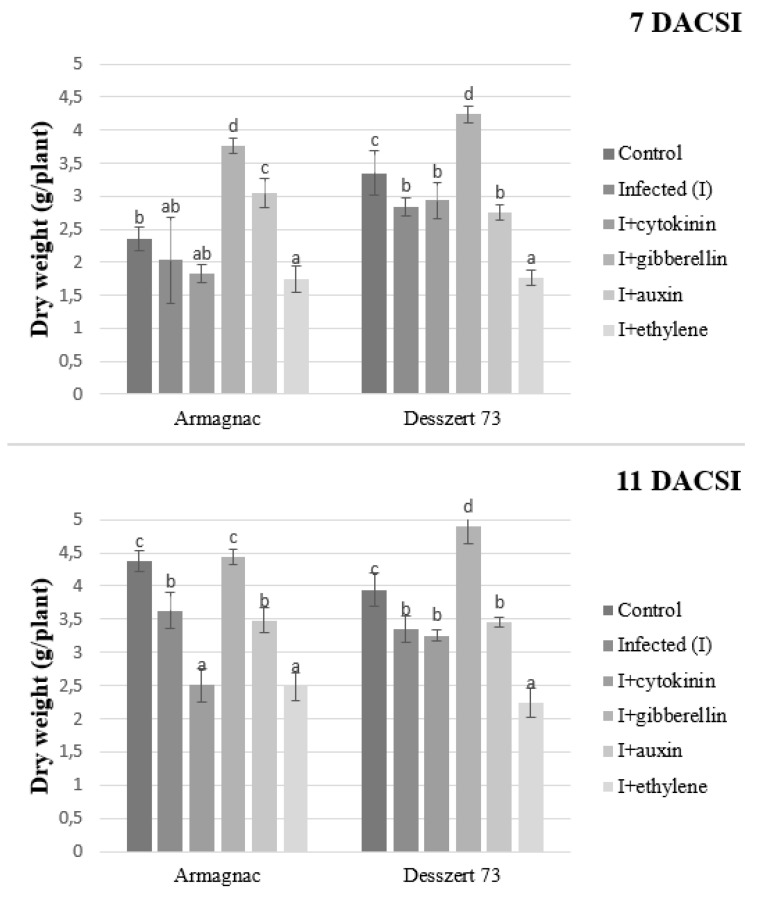
The shoot dry weight (mean ± SD, *n* = 5) of Armagnac and Desszert 73 hybrids (7 and 11 DACSI). Lower case letters (a, b, c, and d) show significant differences among the treatments based on the Tukey HSD test (*p* < 0.05).

**Figure 10 plants-10-01822-f010:**
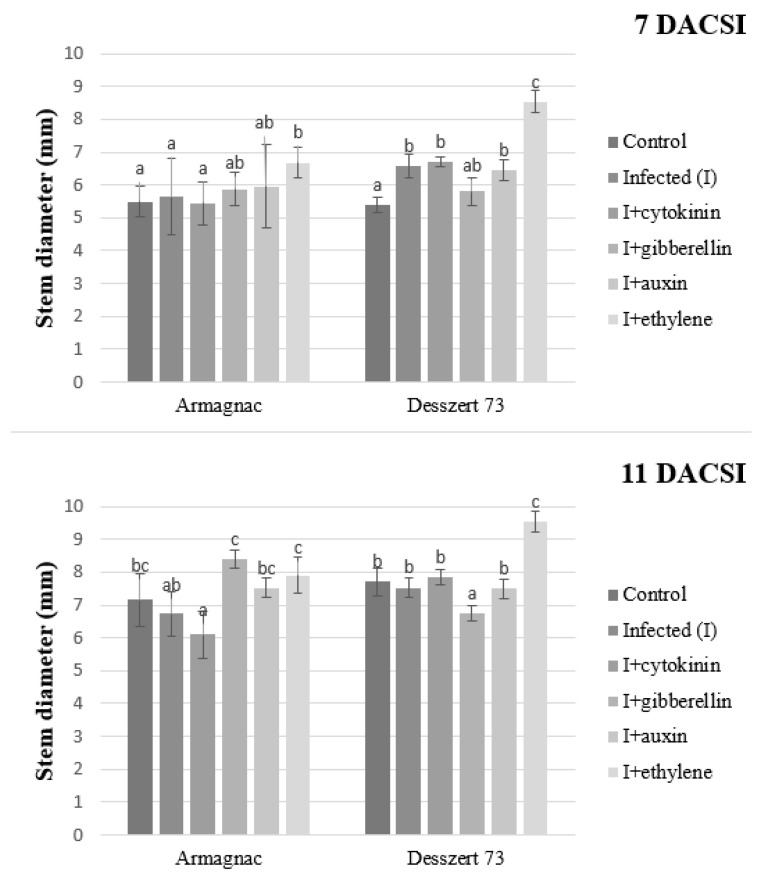
The stem diameter (mean ± SD, *n* = 5) of Armagnac and Desszert 73 hybrids (7 and 11 DACSI). Lower case letters (a, b, and c) show significant differences among the treatments based on the Tukey HSD test (*p* < 0.05).

**Table 1 plants-10-01822-t001:** Average values of two hybrids and two sampling times for measured parameters for six treatments.

	Control Mean ± SD	Infected Mean ± SD	Cytokinin Mean ± SD	Gibberellin Mean ± SD	Auxin Mean ± SD	Ethylene Mean ± SD
Chlorophyll-*a*	15.75 ± 0.55 c	12.58 ± 1.92 b	15.735 ± 2.07 c	12.23 ± 3.49 b	12.175 ± 2.32 b	9.44 ± 3.25 a
Chlorophyll-*b*	5.95 ± 1.29 c	4.49 ± 1.45 b	5.79 ± 0.95 c	5.12 ± 2.64 bc	4.45 ± 2.64 b	3.27 ± 1.72 a
Carotenoid	3.62 ± 3.29 bc	2.97 ± 2.15 ab	3.37 ± 3.42 b	3.88 ± 1.74 c	3.39 ± 1.51 b	2.49 ± 1.09 a
MDA	19.1 ± 3.97 a	72.35 ± 27.91 c	35.68 ± 5.46 b	38.64 ± 12.89 b	34.31 ± 17.22 b	110.23 ± 29.82 d
POD	0.125 ± 0.29 a	0.314 ± 0.12 c	0.136 ± 0.45 a	0.197 ± 0.65 b	0.212 ± 0.48 b	0.365 ± 0.021 c
APX	0.202 ± 0.05 a	0.515 ± 0.11 d	0.227 ± 0.05 b	0.313 ± 0.08 c	0.317 ± 0.11 c	0.592 ± 0.24 e
SOD	0.072 ± 0.001 a	0.105 ± 0.003 c	0.081 ± 0.003 b	0.073 ± 0.004 a	0.082 ± 0.002 b	0.108 ± 0.004 c
Plant height	627 ± 217.74 d	592 ± 246.20 c	559 ± 161.55 b	743 ± 254.19 e	643 ± 258.47 c	438 ± 171.67 a
Stem Diameter	3.45 ± 0.23 bc	2.56 ± 0.58 b	3.40 ± 0.04 bc	2.82 ± 1.50 bc	2.45 ± 1.33 b	1.79 ± 0.99 a
Shoot dry weight	3.51 ± 0.76 c	2.96 ± 0.61 bc	2.63 ± 0.53 b	4.33 ± 0.41 d	3.18 ± 0.31 bc	2.06 ± 0.32 a

Lower case letters (a, b, c, d and e) show significant differences among the treatments line by line based on the Tukey HSD test (*p* < 0.05).

**Table 2 plants-10-01822-t002:** Average values for measured parameters in two hybrids with six treatments over two sampling times.

	Armagnac Mean ± SD	Desszert 73 Mean ± SD	LSD
Chlorophyll-a	12.88 ± 2.99 a	13.09 ± 3.39 a	0.378
Chlorophyll-b	4.43 ± 1.83 b	1.88 ± 1.88 a	0.038
Carotenoids	6.29 ± 1.56 a	5.98 ± 1.70 a	0.121
MDA	28.88 ± 9.07 a	74.56 ± 24.45 b	0.025
POD	0.30 ± 0.11 a	0.37 ± 0.19 a	0.216
APX	0.29 ± 0.13 a	0.43 ± 0.18 b	0.014
SOD	0.0815 ± 0.0006 a	0.08901 ± 0.0018 a	0.189
Plant height	751.00 ± 80.43 b	449.67 ± 96.61 a	0.030
Stem diameter	6.57 ± 0.99 a	7.20 ± 1.16 b	0.036
Shoot dry weight	2.98 ± 0.95 a	3.25 ± 0.85 a	0.176

Lower case letters (a, and b) show significant differences between hybrids line by line based on the Tukey HSD test (*p* < 0.05).

## Data Availability

Data is contained within the article.

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
