# Peer review of "The Application of Phytohormones as Biostimulants in Corn Smut Infected Hungarian Sweet and Fodder Corn Hybrids"

_plants, 2021, doi:10.3390/plants10091822_

Round 1

Reviewer 1 Report

The work has interesting aspects but needs careful revision as the conclusions are not fully supported by the parameters investigated. In particular, it is not clear how the hormones studied could have carried out their protective action. Since this is the central aspect of the work, a few more indications are needed. In addition, the English in some parts needs to be revised. Finally, the manuscript can be shortened because the results section and the discussion section are confusing and repetitive.

Lines 67-68: rephrase the sentence in a clearer form.

Lines 74-75: It would be better to indicate against which ROSs the enzymes mentioned are involved, rather than talking about free radicals.

Lines 111-114: rephrase the sentence. In this form, it is hard to understand.

Figure 5: the content of protein should refer to plant fresh weight and not to mg/L.

Figure 7: the y-axes should report the unit measure as U mg protein-1

All the Figures: Figures for each parameter studied (7 DCASI and 11 DACSI) should be merged into one figure.

Lines 257-265: This part repeats the results and therefore adds nothing new. Furthermore, the effects of hormones on chlorophyll are scarcely supported by the bibliography indicated. Lease, better discuss this last aspect.

Lines 280-282: The effect of hormones on MDA content needs to be better explained and discussed

Line 288: use the acronym, without the full name, also for the ascorbate peroxidase

Line 288-289: The discussion cannot be like this because the authors have not determined hydrogen peroxide content.

Lines 393-394: it would be necessary to indicate why these concentrations of hormones were chosen for the trial conducted.

Line 398: Were the times indicated, i.e. 7 and 11 days after the corn smut infection (DACSI), chosen for specific reasons? Did the authors' previous studies give any indication of this?

The number of replications per treatment is missing in the statistical analysis.

Author Response

The work has interesting aspects but needs careful revision as the conclusions are not fully supported by the parameters investigated. In particular, it is not clear how the hormones studied could have carried out their protective action. Since this is the central aspect of the work, a few more indications are needed. In addition, the English in some parts needs to be revised. Finally, the manuscript can be shortened because the results section and the discussion section are confusing and repetitive.

The authors wish to thank for the valuable comments of the reviewer and have implemented the suggested changes in the manuscript according to the reviewer’s comments. We hope that all the points have been adequately addressed. The changes that have been introduced in the revised manuscript are highlighted in red font, and the answers to the comments are given below.

The manuscript is checked by a native English speaker.

Lines 67-68: rephrase the sentence in a clearer form.

It is corrected in the manuscript. The impacts of oxidative burst induced by Fusarium reduced the protein content in maize leaves.

Lines 74-75: It would be better to indicate against which ROSs the enzymes mentioned are involved, rather than talking about free radicals.

It is corrected in the paper. The selected enzymes in this study are playing a role in the transformation of free radicals into harmless metabolites. Superoxide-dismutase (SOD) transforms superoxide into molecular oxygen and hydrogen peroxide. Ascorbate oxidase (APX) participates in ascorbate metabolism, using L-ascorbate and oxygen to produce dehydroascorbate and water.  Peroxidase (POD) breaks down hydrogen peroxide producing molecular oxygen and water.

Lines 111-114: rephrase the sentence. In this form, it is hard to understand.

It is corrected in the manuscript. Chlorophyll-a concentration was lower in corn smut infected Dessert 73 (7 and 11 DACSI) and Armagnac (11 DACSI) relative to non-infected control. Infected Desszert 73 (11 DACSI) treated with cytokinin, gibberellin, and auxin, had significantly higher chlorophyll-a content compared to infected control.

Figure 5: the content of protein should refer to plant fresh weight and not to mg/L.

The authors apologize for their mistake. The unit of presented protein content was mg/ml. The figure contained the protein content of enzyme extracts which were used for the calculation of enzymes activities, so authors deleted that figure and protein results in this MS.

In order to express the ‘specific enzyme activity’, enzymes should be expressed per protein content not fresh weight.

Figure 7: the y-axes should report the unit measure as U mg protein-1

It is corrected in the manuscript.

All the Figures: Figures for each parameter studied (7 DCASI and 11 DACSI) should be merged into one figure.

It is corrected in the manuscript.

Lines 257-265: This part repeats the results and therefore adds nothing new. Furthermore, the effects of hormones on chlorophyll are scarcely supported by the bibliography indicated. Lease, better discuss this last aspect.

Based on the reviewer suggestion this part was deleted, and following information is added:

Cytokinins play a role in chlorophyll synthesis, chloroplast ultrastructure, and chloroplast differentiation (Cortleven and Schülling, 2015). Dobránszki and Mendler-Drienyovszki (2014) stated that cytokinin treatment has impact on chlorophyll a and b content in vitro apple leaves. Its effect depends on the applied concentration and form of cytokinin. Additionally, the application of phenylacetic and α-naphtylacetic acid reduced the chlorophyll and caroteinoid contents in Wolffia arrhiza (Czerpak et al., 2002). Research studies represented that indol-acetic acid (IAA) and naphthaleneacetic acid (NAA) treatments generated chlorophyll loss in tobacco (Even-Chen et al., 1978), and in lettuce (Aharoni, 1989). Similar statements could be found related to ethylene. Ceusters and Van de Poel (2018) published that the application of ethylene causes the degradation of chlorophyll through the induced chlorosis.

Lines 280-282: The effect of hormones on MDA content needs to be better explained and discussed

The following information is added to the MS:

Phytohormones play a role in the stimulation of fatty acid biosynthesis, mitigate the negative effects of oxidative burst, and increase the lipid production (Du et al., 2017; Lu and Xu, 2015).

Sivaramakrishnan and Incharoernsakdi (2020) noticed that MDA content was significantly higher after indole acetic acid and gibberellic acid (GA) treatments in Chlorella sp. Higher MDA content was observed at IAA relative to GA with the concentration of 1mM. They concluded that the application of IAA induces the generation of free radicals, which cause higher MDA content and cell damage.

Line 288: use the acronym, without the full name, also for the ascorbate peroxidase

The authors corrected this in the manuscript. But, we disagree with this correction because we think not customary to start a sentence with an acronym.

Line 288-289: The discussion cannot be like this because the authors have not determined hydrogen peroxide content.

Authors think, the reviewer mentioned Line 320-321. In this experiment, maize plants tried to decrease the H2O2 and ROS accumulation by increasing their enzymes activities after corn smut infection.”  So, that sentence is corrected to the following:  In this experiment, maize plants reacted to corn smut infection in order to decrease the free radical accumulation by increasing their enzymes’ activities.

Lines 393-394: it would be necessary to indicate why these concentrations of hormones were chosen for the trial conducted.

The selected hormones concentrations are higher than in plants because we wanted to examine the hormones’ effects, so there concentrations are provocative. Authors added this information at the end of the introduction.

Line 398: Were the times indicated, i.e. 7 and 11 days after the corn smut infection (DACSI), chosen for specific reasons? Did the authors' previous studies give any indication of this?

Based on our previous experiment, which was conducted to observe the time of first symptoms of the CSI, the first symptoms (chlorosis, and tumor formation) of corn smut infection appeared 7 DACSI. This was the reason for the first sampling time. The second sampling time (11 DACSI) was chosen because sweet corn started to get necrosis symptoms 14 DACSI. We included this information at the end of the introduction.

The number of replications per treatment is missing in the statistical analysis.

The number of replications was five per treatment for statistical analysis. We added this information to the manuscript.

Reviewer 2 Report

The results do not call for the attention of the readers. The introduction is a bit long and would it be better if it were re-written. Bibliography is sound and diverse.

Author Response

The results do not call for the attention of the readers. The introduction is a bit long and would it be better if it were re-written. Bibliography is sound and diverse.

The authors wish to thank for the valuable comments of the reviewer and have implemented the suggested changes in the manuscript according to the reviewer’s comments. We hope that all the points have been adequately addressed. The changes that have been introduced in the revised manuscript are highlighted in red font, and the answers to the comments are given below.

The authors rewrote the Introduction of the manuscript but because of the other reviewers’ suggestions it is longer in the present form.  

Reviewer 3 Report

Dear Authors and Editors,

The manuscript submitted for review in its current form is not ready for publication. I am asking the authors to read carefully my detailed comments, which I have included in the file (please see below). The research is interesting, but from the biochemical-physiological and phytopathological points of view, the manuscript still requires corrections and improvements. I will be very grateful for your response to my comments. In my opinion, after their introduction, this manuscript will be of better quality and will be ready to send out for the compulsory next round of reviews.

Sincerely

Reviewer

Author Response

The manuscript submitted for review in its current form is not ready for publication. I am asking the authors to read carefully my detailed comments, which I have included in the file (please see below). The research is interesting, but from the biochemical-physiological and phytopathological points of view, the manuscript still requires corrections and improvements. I will be very grateful for your response to my comments. In my opinion, after their introduction, this manuscript will be of better quality and will be ready to send out for the compulsory next round of reviews.

The authors wish to thank for the valuable comments of the reviewer and have implemented the suggested changes in the manuscript according to the reviewer’s comments. We hope that all the points have been adequately addressed. The changes that have been introduced in the revised manuscript are highlighted in red font, and the answers to the comments are given below.

Please change to Ostrinia nubilalis

It is corrected in the manuscript.

Please change to Mythimna separate

It is corrected in the manuscript.

Please change to radiate

It is corrected in the manuscript.

To my knowledge, there are information in the literature regarding the biochemical-physiological interaction of the host plant with Ustilago maydis. Particular attention should be paid to the issues of free radicals, oxidative stress, oxidative damage and hormone balance during interactions. I am asking the authors to carefully study the literature on the issues mentioned by me and introduce.

If the reviewer is particular related to some special publications, please let the authors know.

The following information is added to the MS:

Ustilago maydis can cause oxidative damage in host plants through the enhanced free radical generation (Lambie et al., 2017).  Hemetsberger et al. (2012) reported that corn smut has a specific protein, called Pep1, which can block the activity of peroxidase12 enzyme, resulting the inhibition of oxidative burst as a respond of host to smut infection.

Pathogens activate the biosynthesis of hormones in plants (Choi et al., 2011). Corn smut is a great example the host plant’s hormone balance manipulation, as a biothrope fungus. Bölker et al. (2008) noticed that the confused hormone balance of host caused by corn smut infection has a role in host infection. Research data shows that corn smut able to produced cytokinins, abscisic acid (Morrison et al., 2015), and auxins (Reineke et al., 2008) to manipulate the host’s hormone balance. Cytokinins and abscisic acid also play a role in tumor formation (Bruce et al., 2011).

In my opinion, the introduction should introduce basic information about these hormones. As things stand, information appears only in the section on research goals. I ask the authors to briefly present the role of these hormones in biotic stress in a physiological way and then clearly outline the purpose for which it was decided to treat with such hormones.

The following information is added to the Introduction section:

The main phytohormones are auxins, gibberellins, cytokinins (CK), abscisic acid (ABA), and ethylene (ET), salicylic acid (SA), jasmonates (JA), brassinosteroids (BR) and strigolactones. The role of phytohormones like ethylene, jasmonic acid, and salicyclic acid are widely examined in response to plants’ immune system to biotic stresses (Dong, 1998; Díaz et al., 2002; van Loon et al., 2006; Yang et al., 2016; Xiong et al., 2020). Additionally, auxins, gibberellins, and cytokinins are also described as resistance regulators (Pieterse, et al., 2012; Denancé et al., 2013; Hurny et al., 2020).  Denancé et al. (2013) stated that the application of phytohormones can enhance the resistance of host plants under biotic stress conditions. Manipulation of phytohormes homeostasis in crops can lead to a better resistance response to biotic factors.  Auxins plays a role especially in growth regulation (Schepetilnikov and Ryabova, 2017). In addition, auxins are also important in the regulation of pathogen resistance (Fu and Wang, 2011), e.g. auxin-asparted acid played a role in the resistance to a necrotrophic fungus, Botrytis cinerea (Denancé et al., 2013). Several studies have dealt with the role of ethylene in pathogen resistance. These found that the application of ethylene enhances the resistance or susceptibility of host plants. The effect depends on the type of pathogen and host plants (Esquerré Tugayé et al., 1979; van Loon and Pennings, 1993). Cytokinins are important in cell division and differentiation (Schaller et al., 2014), chloroplast structure, senescence (Hönig et al., 2018), and in the reaction of host plants to biotic stressors (Sakakibara, 2006; Akhtar et al., 2020).  Gibberellins can enhance or decrease the host plants’ resistance to phytopathogens, they play a role especially in the respond of host plants to biotrophic and necrotrophic pathogens (Bari and Jones, 2009).

Why was the protein content per liter expressed? This is not correct, please change it by expressing by gram of tissue. This remark applies to all units, please standardize it so that the case of MDA, we see the content per g of FW, while photosynthetic pigments per g, but without specifying whether it was FW, or for example, DW.

The authors apologize for their mistake. The unit of presented protein content was mg/ml. That figures contained the protein content of enzyme extracts what were used for the calculation of enzymes activities, so authors deleted that figures and protein results in this MS.  Photosynthetic pigments were measured based on fresh weight. We changed it in the MS.

As the protein content was variable in your experiment, perhaps from a methodical point of view it would be better to express the enzyme activity per g of tissue. What do you think?

Because the protein content of the enzymes’ extracts is varied among the treatments, authors think that this presentation is more precise than the weight unit. It is important to express the specific enzyme activity per “mg” protein not per fresh weight.

Please recalculate all enzymes activities in the same way. Currently we see different units, this is not correct.

The activity of SOD was recalculated based on protein content (enzyme units mg-1 protein).

My big concern about the manuscript is the lack of all the controls. In my opinion, there is a lack of control here: uninfected control supplemented with phytohormones (cytokinin, gibberellin, auxin, and ethylene). Without these controls it is difficult to draw methodically and metrically correct conclusions. I am asking the authors to address this very important matter. The authors do not agree that the experiment should have had a non-infected and phytohormones controls. The effects of phytohormones on plant growth and development are well defined in the literature. Students are learning this during their university studies. The aim of this experiment was to examine the effect of corn smut infection on the host plants’ measured characteristics. To complete this goal, the results of the infected plants were relative to the non-infected plants, as a negative control. The second aim of this study was to evaluate the impacts of four (auxin, gibberellin, cytokinin, and ethylen) phytohormones under corn smut infection. To achieve this goal the results of hormones treated and infected plants were compared to the infected plants’ results, as a positive control. If we would have applied “non-infected plants treated with phytohormones treatment” that would have meant that we examine the effect of phytohormones under “normal”, not stressed conditions.  The aim of this study was not that.

Why were such measuring points chosen? Is it related to the biology of the pathogen? Please also clarify this and it is best to mention it in the introduction (at the end).

Based on our previous experiment, which was conducted to observe the time of first symptoms of the CSI, the first symptoms (chlorosis, and tumor formation) of corn smut infection appeared 7 DACSI. This was the reason for the first sampling time. The second sampling time (11 DACSI) was chosen because sweet corn started to get necrosis symptoms 14 DACSI. We put this information at the end of the introduction.

Round 2

Reviewer 1 Report

The paper can be accepted 

Author Response

The authors thank for the reviewer’s time for checking the revised version of their manuscript. In addition, thanks a lot that they accepted our responses for their questions.

Reviewer 2 Report

The results are better explained. The introduction has increased and I believe it has too much information, so it should be reduced.

Author Response

The authors thank for the reviewer’s time for checking the revised version of their manuscript.

The Introduction was extended based on reviewer 1 and reviewer 3’s request. The authors deleted some information in the Introduction part, but they did not want to delete the requested information based on other reviewers’ comments. If the reviewer is specific about the unnecessary part(s) in the introduction, please let the authors know.

Reviewer 3 Report

See file below.

Author Response

The authors thank for the reviewer’s time for checking the revised version of their manuscript.